# Prediction of radiation pneumonitis using dose-volume histogram parameters with high attenuation in two types of cancer: A retrospective study

Yasuki Uchida[1], Takuya Tsugawa[2], Sachiko Tanaka-Mizuno[3,4], Kazuo Noma[5], Ken Aoki[2], Kentaro Fukunaga[1], Hiroaki Nakagawa[1], Daisuke Kinose[1], Masafumi Yamaguchi[1], Makoto Osawa[1,6], Taishi Nagao[1], Emiko Ogawa[1,7], Yasutaka Nakano[1]*

1 Division of Respiratory Medicine, Department of Internal Medicine, Shiga University of Medical Science, Otsu, Shiga, Japan, 2 Department of Radiology, Shiga University of Medical Science, Otsu, Shiga, Japan, 3 Department of Medical Statistics, Shiga University of Medical Science, Otsu, Shiga, Japan, 4 The Center for Data Science Education and Research, Shiga University, Hikone, Shiga, Japan, 5 Department of Radiology, Shiga University of Medical Science Hospital, Otsu, Shiga, Japan, 6 Division of Infection Control and Prevention, Shiga University of Medical Science Hospital, Otsu, Shiga, Japan, 7 Health Administration Center, Shiga University of Medical Science, Otsu, Shiga, Japan

* nakano@belle.shiga-med.ac.jp

**Data Availability Statement:** All relevant data are within the manuscript and its Supporting Information files.

## Abstract

The constraint values of dose-volume histogram (DVH) parameters for radiation pneumonitis (RP) prediction have not been uniform in previous studies. We compared the differences between conventional DVH parameters and DVH parameters with high attenuation volume (HAV) in CT imaging in both esophageal cancer and lung cancer patients to determine the most suitable DVH parameters in predicting RP onset. Seventy-seven and 72 patients who underwent radiation therapy for lung cancer and esophageal cancer, respectively, were retrospectively assessed. RP was valued according to the Common Terminology Criteria for Adverse Events. We quantified HAV with quantitative computed tomography analysis. We compared conventional DVH parameters and DVH parameters with HAV in both groups of patients. Then, the thresholds of DVH parameters that predicted symptomatic RP and the differences in threshold of DVH parameters between lung cancer and esophageal cancer patient groups were compared. The predictive performance of DVH parameters for symptomatic RP was compared using the area under the receiver operating characteristic curve. Mean lung dose, HAV30% (the proportion of the lung with HAV receiving $\geq$30 Gy), and HAV20% were the top three parameters in lung cancer, while HAV10%, HAV5%, and V10 (the percentage of lung volume receiving 10 Gy or more) were the top three in esophageal cancer. By comparing the differences in the threshold for parameters predicting RP between the two cancers, we saw that HAV30% retained the same value in both cancers. DVH parameters with HAV showed narrow differences in the threshold between the two cancer patient groups compared to conventional DVH parameters. DVH parameters with HAV may have higher commonality than conventional DVH parameters in both patient groups tested.

**Funding:** This work was supported by Daiichi-Sankyo Company, Limited [grant number A19-1287], (YU), https://www.daiichisankyo.co.jp/corporate/ds-shougakukifu/, Ono Pharmaceutical Company, Limited [grant number ONOS20180618011], (NY), https://kifu-shinsei.jp/kifu-entry/cmn/doc/index_N1gsZhJaOt.html, Pfizer Inc [grant number 54334665], (NY), https://pfizer-ac-web.pfizer.co.jp/detail.html, Bayer Yakuhin, Limited [BASJ20180409030], https://byl.bayer.co.jp/researchers/, Boehringer Ingelheim GmbH [grant number RS2019A00769379], (NY), https://www.boehringer-ingelheim.jp/Research_support_2019, and Astelas Pharma [grant number RS2019A001313], (NY), https://www.astellas.com/jp/ja/responsibility/astellas-foundations, and had no role in study design, data collection, data analysis, data interpretation, or writing of the report.

**Competing interests:** Daiichi-Sankyo Company, Limited, Ono Pharmaceutical Company, Limited, Pfizer Japan, Bayer Corporation, Boehringer Ingelheim GmbH, and Astelas Pharma Global Development funded this study and had no role in study design, data collection and analysis, decision to publish, or preparation of the manuscript. The authors received no specific funding for this work and declare that they have no competing interests. This does not alter our adherence to PLOS ONE policies on sharing data and materials.

**Abbreviations:** RP, Radiation pneumonitis; DVH, dose-volume histogram; HAV, high attenuation volume; HAVx%, the proportion of the lung with HAV receiving x Gy or more; MLD, mean lung dose; Vx, the percentage of lung volume receiving x Gy or more; CT, computed tomograph; LAV, Low attenuation volume; TLV, total lung volume; ILD, interstitial lung disease; AUC, the area under the receiver operating characteristic curve; COPD, chronic obstructive pulmonary disease; MHALD, mean high attenuation lung dose.

# Introduction

Radiation pneumonitis (RP) is a serious adverse effect of radiation therapy; symptomatic pneumonitis particularly negatively influences chemotherapy after radiotherapy. Therefore, the prediction and prevention of symptomatic RP are very significant. Even among lung cancer patients alone, various threshold values of DVH parameters have been reported for the prediction of RP [1–3]. Thus, guidelines for thoracic radiotherapy have set constraint values for traditional DVH parameters such as the mean lung dose (MLD) (20–23 Gy) and the percentage of lung volume receiving 20 Gy or more (V20) (30–40%) to reduce the risk of RP [4–8]. We recently showed that DVH parameters calculated using only volumes of high attenuation in CT imaging by excluding emphysematous lesions were better predictors of RP than traditional parameters [9]. However, little has been reported on the threshold values of DVH parameters for the prediction of RP in different types of cancers. Threshold values that are closer or the same in different types of cancers would be more reliable and versatile. We compared the performances of our new DVH parameters with traditional parameters in predicting RP in both esophageal cancer patients and lung cancer patients to evaluate whether our new DVH parameters could predict RP more accurately, without varying threshold values among different types of cancer.

# Methods and materials

## Study design

This study was a single-center, retrospective, observational study. The endpoint of this analysis was to evaluate the performance of DVH parameters for the prediction of symptomatic RP of grade 2 or worse (Common Terminology Criteria for Adverse Events, version 5.0).

## Selection of study participants

Patients who received radiotherapy for lung cancer (n = 77) or esophageal cancer (n = 72) at our institution between June 2010 and July 2017 were selected retrospectively. The inclusion criteria were as follows: first time receiving radiotherapy, total irradiation dose >30 Gy (fraction dose is 1.8–3.0 Gy), pneumonectomy not performed within 5 months after radiotherapy or before the occurrence of symptomatic RP, follow-up period >5 months if symptomatic RP did not occur, and entire lung fields scanned using computed tomography (CT) before radiotherapy. We excluded patients who underwent stereotactic body radiotherapy. Patients were treated with either curative or palliative intent with radiotherapy alone or with concurrent chemoradiation.

## Radiotherapy planning and image analysis

Radiotherapy planning was performed as 3D treatment planning using the Eclipse$^{TM}$ software (Varian Medical Systems, Palo Alto, CA, USA) with an analytical anisotropic algorithm, and the calculation grid was 2.5 mm for the lung. The treatment planning was based on a 1.25-mm thick CT scans obtained in the treatment position. The resolution of CT scans was 1.25 mm×1.25 mm (field of view = 64 cm, matrix = 512×512 pixels). The distribution of the radiation dose was calculated using lung heterogeneity corrections. The breathing phase of the CT scan was free breathing. Gating and breath-hold were not used. We evaluated DVH parameters without emphysematous lesions, as previously described [5, 9]. Low attenuation volume (LAV), which expresses emphysematous lesions in the lung, was assessed using the upper threshold limit of −856 HU. To further verify the validity of this constraint value, we previously validated the association between the CT under free-breathing and the inspiratory CT

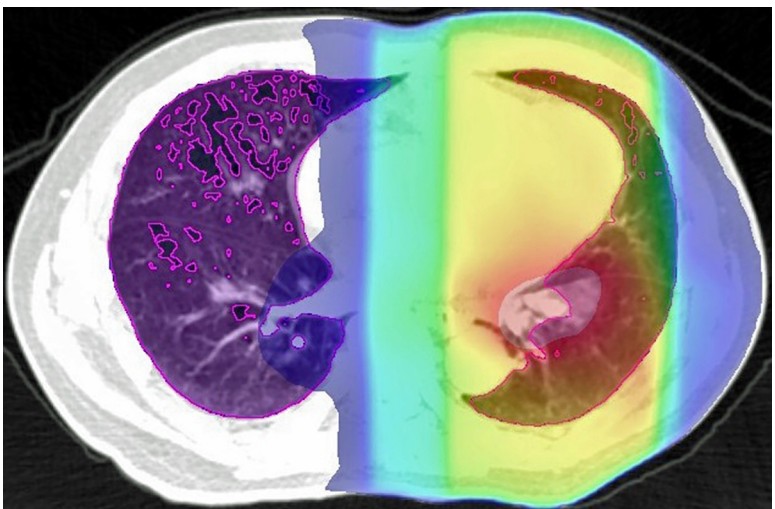

**Fig 1. Area inside the purple line shows the high attenuation lung using a threshold of −856 HU.** The colorful area represents the irradiated area (red indicates the area with the highest dose, and blue represents the area with the lowest dose), and the overlaps were calculated.

performed within 45 days after free-breathing CT [10]. Inspiratory CT was performed using Toshiba Aquillion ONE (Toshiba Medical Systems Corp., Otawara, Tochigi, Japan), and LAV was analyzed using Aquarius iNtuitionTM software ver.4.4.12 (TeraRecon Inc., San Mateo, Calif) and evaluated using the threshold limit of -950 HU. The LAV in inspiratory CT was highly correlated with the LAV in CT under free breathing. Regarding relative electron density, −856HU was 0.1488. Since "total lung volume (TLV)–LAV" is equal to high attenuation volume (HAV) ≥−856 HU in the lung, we described "TLV–LAV" as HAV in this report (Fig 1). We also defined HAV irradiated at ≥30 Gy as HAV30 and "HAV30 / TLV" as HAV30%. We evaluated the DVH parameters that predicted RP more accurately in our previous study (See S1 Table). The mean high attenuation lung dose (MHALD) was defined as the mean dose irradiated on a high attenuation lung field.

## Clinical toxicity

The severity of RP was assessed retrospectively using the Common Terminology Criteria for Adverse Events, version 5.0 [11]. Patients were generally followed up for 3 to 6 weeks after the completion of radiotherapy, and at 3- to 6-month intervals thereafter. The diagnosis of RP and the evaluation of its severity was performed based on radiographic images, laboratory test results, physical examination findings, and clinical symptoms, by reviewing medical records.

The study protocol was approved by the Institutional Review Board of the Shiga University of Medical Science (IRB no. R2014-236).

## Statistical analyses

We used summary statistics to analyze clinical factors such as age, sex, disease stage, histologic type, type of radiotherapy methods, chemotherapy, smoking history, smoking index, body mass index, and interstitial lung disease (ILD) for all patients. Continuous variables are presented as medians and ranges, and categorical variables as percentages. We also compared clinical factors between RP ≥grade 2 and RP ≤grade 1 using Wilcoxon's rank-sum test or Fisher's exact test, as appropriate.

Conventional DVH parameters, including MLD, V2, V5, V10, V20, and V30, and other DVH parameters of high lung attenuation, are described as medians and interquartile ranges.

Univariate logistic regression analysis was performed to evaluate the association between each DVH parameter and the onset of symptomatic RP. The predictive performance of each DVH parameter for RP was compared using the area under the receiver operating characteristic curve (AUC). We identified the optimal decision threshold value of each DVH parameter with the highest sensitivity and specificity. Statistical analyses were performed using JMP version 11 (SAS Institute Inc., Cary, NC, USA) and R version 3.5.1 (R Foundation for Statistical Computing, Vienna, Austria) [12]. Analyzed items with $P<0.05$ were considered statistically significant.

## Results

### Clinical parameters

RP was observed in 43 out of 77 lung cancer patients (grade 1, n = 14; grade 2, n = 13; grade 3, n = 13; grade 4, n = 1; and grade 5, n = 2) and in 27 out of 72 esophageal cancer patients (grade 1, n = 19; grade 2, n = 3; grade 3, n = 4; grade 4, n = 0; and grade 5, n = 1).

In the univariate analysis of lung cancer patients, concurrent chemotherapy, pre-existing ILD, MLD, and V20 were significantly correlated with the occurrence of symptomatic RP (Table 1). In the univariate analysis of esophageal cancer patients, smoking history, MLD, and V20 were significantly correlated with the occurrence of symptomatic RP (Table 2). Chemotherapeutic agents are summarized in S2 Table.

### DVH parameters

In both lung cancer and esophageal cancer patients, univariate logistic regression analysis for symptomatic RP ($\geq$grade 2) showed that all DVH parameters were significantly related to symptomatic RP (See S3 Table). When the predictive performances of DVH parameters for symptomatic RP were compared using the AUC, MLD, HAV30%, and HAV20% were the three best parameters in lung cancer and HAV10%, HAV5%, and V10 were the three best in esophageal cancer (Fig 2). As there were no overlaps between lung cancer and esophageal cancer, we compared the thresholds of these parameters for the prediction of RP between the two forms of cancer (Fig 3). When the differences in threshold of parameters between the two cancers were compared, threshold values of HAV30% were found to be almost identical in these cancers. For all DVH parameters, differences in the threshold between the two cancers were smaller when considering non-emphysematous (MLHAD, HAV30%, HAV20%, HAV10%, and HAV5%) than conventional (MLD, V30, V20, V10, and V5) parameters (Fig 4). Boxplots for each DVH parameter are shown in S1 Fig.

## Discussion

V20 or MLD has been commonly used as an index for the prevention of severe RP in practice. In various clinical applications of radiotherapy, the limit of MLD or V20 has been described [13–16]. These DVH parameters are simple to calculate and accurate enough for the prediction of RP but have never been compared with other DVH parameters in this regard. In addition, the reported threshold values of dose-volume histograms for the prediction of RP have only been discussed for lung cancer patients [1–3]. We previously showed that DVH parameters representing irradiated non-emphysematous lung volume were better predictors of RP than conventional DVH parameters [9]. Little has been reported on the comparison of thresholds of DVH parameters associated with the onset of RP among different malignant populations.

**Table 1. Clinical parameters in lung cancer patients with symptomatic and asymptomatic radiation pneumonitis.**

| Characteristic | Symptomatic Patients (N = 29) | Asymptomatic Patients (N = 48) | P-value |
|---|---|---|---|
| Median age (range), year | 69 (57–82) | 67 (39–89) | 0.678 |
| Male sex | 26 (89.6) | 40 (83.3) | 0.520 |
| Disease stage | | | 0.104 |
| 1 | 0 (0) | 5 (10.4) | |
| 2 | 2 (6.9) | 3 (6.3) | |
| 3 | 24 (82.8) | 29 (60.4) | |
| 4 | 3 (10.3) | 11 (22.9) | |
| *Histology type | | | 0.494 |
| SqCC | 13 (44.8) | 15 (31.2) | |
| Adenocarcinoma | 7 (24.1) | 13 (27.1) | |
| SCC | 7 (24.1) | 8 (16.7) | |
| NSCC | 2 (6.9) | 6 (12.5) | |
| Unknown | 0 (0) | 4 (8.3) | |
| Others | 0 (0) | 2 (4.2) | |
| Treatment type | | | 1.000 |
| IMRT | 1 (3.5) | 3 (6.3) | |
| 3D Conformal | 28 (96.6) | 45 (93.8) | |
| Chemotherapy | | | 0.002 |
| Yes | 23 (79.3) | 20 (41.7) | |
| No | 6 (20.7) | 28 (58.3) | |
| Smoking history | | | 0.561 |
| Current | 12 (41.4) | 14 (29.2) | |
| Former | 14 (48.2) | 28 (58.3) | |
| Never | 3 (10.3) | 6 (12.5) | |
| Smoking (range), pack-years | 45 (0–120) | 42 (0–180) | 0.709 |
| Median BMI (range), kg/m$^2$ | 20.55 (16.19–24.83) | 19.47 (14.98–25.42) | 0.091 |
| ILD | | | 0.004 |
| Yes | 4 (13.8) | 0 (0) | |
| No | 25 (86.2) | 48 (100) | |
| Median MLD (IQR), Gy | 13.307 (8.89–17.006) | 7.056 (4.123–9.367) | <0.0001 |
| Median V20 (IQR) | 23.227 (18.047–32.352) | 13.554 (7.526–14.423) | <0.0001 |
| Median LAV% (IQR) | 8.5 (3.6–24.3) | 11.1 (2.6–28.5) | 0.801 |

*Percentages in this column may not add up to exactly 100% because of rounding.

Unless otherwise specified, data are expressed as numbers of patients, and numbers in parentheses are percentages. RP = radiation pneumonitis; SqCC = squamous cell carcinoma; SCC = small cell carcinoma; NSCC = non-small cell carcinoma; IMRT = intensity-modulated radiation therapy; BMI = body mass index; ILD = interstitial lung disease; MLD = mean lung dose; IQR = interquartile range; V20 = percentage of lung volume irradiated $\geq$ 20 Gy; LAV% = ratio of low attenuation volume to the lung volume.

In this study, we assessed the performance of conventional DVH parameters and some DVH parameters calculated by excluding emphysematous lesions reported in our previous study and compared the accuracy using the AUC in esophageal cancer and lung cancer populations. We found that in each population, DVH parameters with HAV predicted the onset of symptomatic RP more accurately than traditional DVH parameters. In addition, the differences in the DVH parameters of the two populations with HAV were smaller than those of traditional DVH parameters. Due to the identical threshold values of HAV30% in these two cancer types, this one threshold value could be used in common, at least in lung cancer and esophageal

**Table 2. Clinical parameters in symptomatic radiation pneumonitis patients and asymptomatic esophageal cancer patients.**

| Characteristic | Symptomatic Patients (N = 8) | Asymptomatic Patients (N = 64) | P-value |
|---|---|---|---|
| Median age (range), year | 69.5 (64–80) | 71 (48–89) | 0.907 |
| Male sex | 7 (87.5) | 56 (87.5) | 1.000 |
| Disease stage | | | 0.905 |
| 0 | 0 (0) | 1 (1.6) | |
| 1 | 1 (12.5) | 7 (10.9) | |
| 2 | 0 (0) | 8 (12.5) | |
| 3 | 4 (50.0) | 30 (46.9) | |
| 4 | 3 (37.5) | 18 (28.1) | |
| *Histology type | | | 0.325 |
| SqCC | 7 (87.5) | 59 (92.2) | |
| Adenocarcinoma | 0 (0) | 3 (4.7) | |
| Unknown | 1 (12.5) | 1 (1.6) | |
| Others | 0 (0) | 1 (1.6) | |
| Chemotherapy | | | 0.585 |
| Yes | 8 (100) | 56 (87.5) | |
| No | 0 (0) | 8 (12.5) | |
| Smoking history | | | 0.036 |
| Current | 0 (0) | 22 (34.4) | |
| Former | 8 (100) | 34 (53.1) | |
| Never | 0 (0) | 8 (12.5) | |
| Smoking (range), pack-years | 44.5 (35–60) | 39 (0–159) | 0.337 |
| Median BMI (range), kg/m$^2$ | 21.99 (15.63–25.37) | 19.59 (14.19–28.56) | 0.282 |
| ILD | | | 1.000 |
| Yes | 1 (12.5) | 9 (14.1) | |
| No | 7 (87.5) | 55 (85.9) | |
| Median MLD (IQR), Gy | 14.43 (10.039–19.258) | 9.637 (7.096–11.884) | 0.007 |
| Median V20 (IQR) | 26.868 (15.349–37.984) | 14.911 (10.150–24.238) | 0.029 |
| Median LAV% (IQR) | 4.6 (0.598–9.032) | 10.781 (2.989–22.868) | 0.082 |

*Percentages in this column may not add up to exactly 100% because of rounding.

Unless otherwise specified, data are expressed as numbers of patients, and numbers in parentheses are percentages. RP = radiation pneumonitis; SqCC = squamous cell carcinoma; SCC = small cell carcinoma; NSCC = non-small cell carcinoma; BMI = body mass index; ILD = interstitial lung disease; MLD = mean lung dose; IQR = interquartile range; V20 = percentage of lung volume irradiated ≥ 20 Gy; LAV% = ratio of low attenuation volume to the lung volume.

cancer. Based on the results presented in this study, we cannot affirm that HAV30% is the best predictor of RP. Indeed, HAV30% was not among the three parameters when the predictive performances of DVH parameters for symptomatic RP were compared using the AUC in esophageal cancer. Moreover, we cannot compare between MLD, MLHAD, and HAV30%, due to the unit differences. However, HAV might be a better predictor than the traditional DVH parameters. The frequency of RP was different in the two groups, but the threshold values were not so different between the groups. This may be because fewer patients with esophageal cancer received irradiation at high doses than did lung cancer patients. Some studies showed that chronic obstructive pulmonary disease (COPD) is a risk factor for RP [17–20], while others showed that RP was less severe in patients with more serious COPD than in patients with normal lung function [21, 22]. Our study is in line with the latter. Some previous studies assessing risk factors for RP [9, 23, 24] showed that emphysematous lesions decreased the risk of RP, which is in line with our results. While we used high attenuation area in the

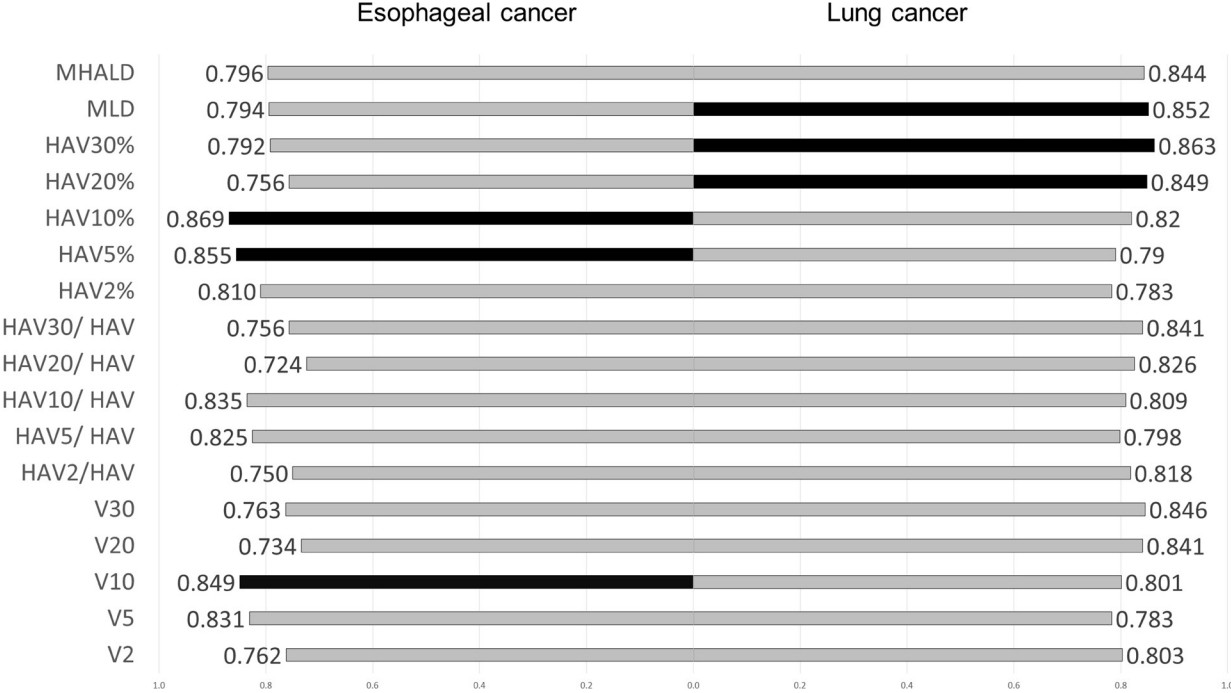

**Fig 2. Comparison of the AUC of dose-volume histogram parameters and RP in lung cancer and esophageal cancer.** Black bars indicate the three best DVH parameters. MLD, HAV30%, and HAV20% were the best three parameters in lung cancer, and HAV10%, HAV5%, and V10 were the three best in esophageal cancer.

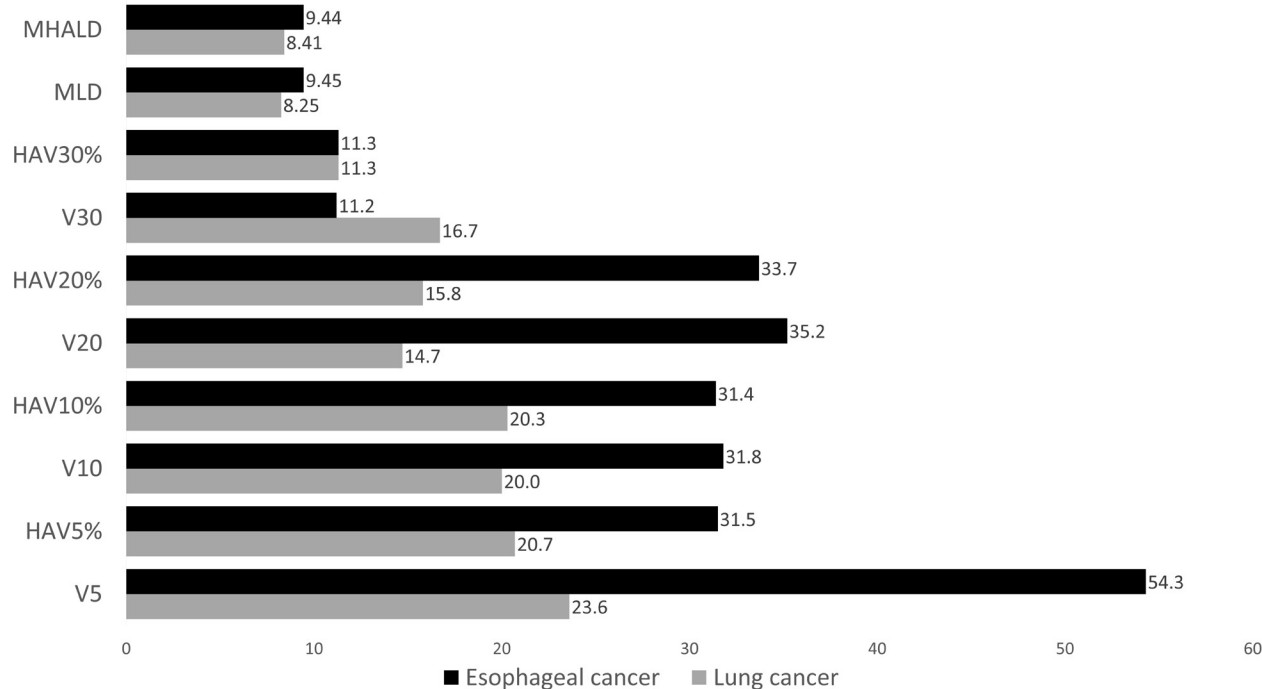

**Fig 3. Threshold of each DVH parameter for the prediction of symptomatic RP.** Black bars represent esophageal cancer, and gray bars represent lung cancer. The same scores were obtained for HAV30% in both esophageal cancer and lung cancer.

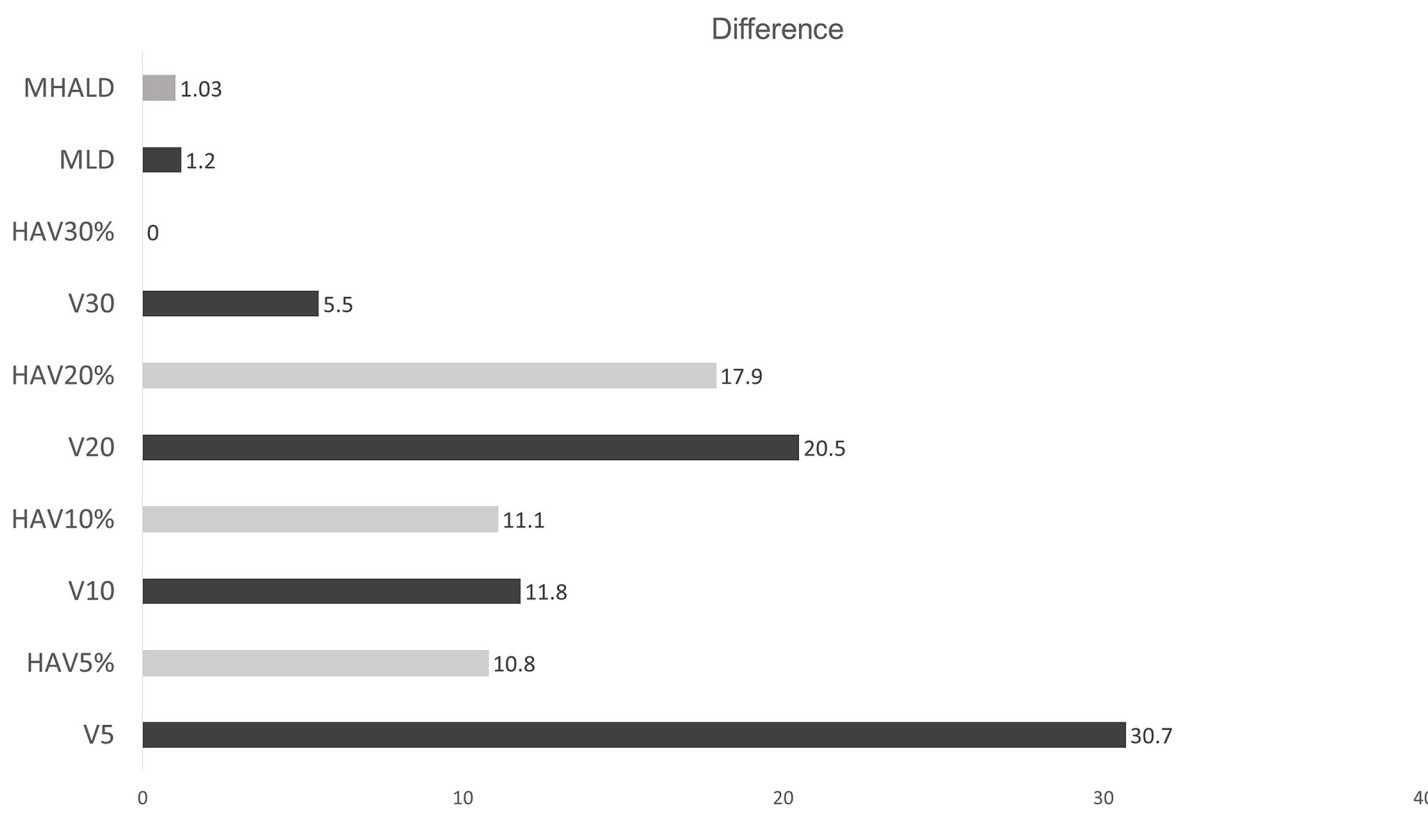

**Fig 4. Comparison of the differences in threshold of each DVH parameter predicting symptomatic RP between lung cancer and esophageal cancer.** Gray bars indicate DVH parameters associated with high attenuation lung volume, and black bars indicate traditional DVH parameters. All DVH parameters associated with high attenuation lung volume (gray bars) were smaller than their counterpart traditional DVH parameters (black bars). The difference in HAV30% was zero.

present study, these previous studies considered the low attenuation area; these have almost the same meaning because the threshold limit of CT value (−856 HU) is the same. The measurement of dose volume using high attenuation area is a very easy and convenient way to predict RP and may be widely applicable. In clinical trials, traditional DVH parameters excluding high attenuation from consideration were used as the standard to avoid RP without doubting their predictive accuracy for RP. To make every possible effort to avoid RP, DVH parameters with high attenuation might be used better when treating lung cancer and esophageal cancer patients.

We used −856 HU as the threshold [10, 25–28]. CT scans were performed under free breathing, meaning they are almost equal to expiratory CT scans. We previously validated that the LAV in inspiratory CT was highly associated with the LAV in CT under free breathing [9].

In our study, four lung cancer patients were treated with intensity-modulated radiotherapy. The type of irradiation used for treatment was not significantly associated with the onset of RP. Whether intensity-modulated radiotherapy increases or decreases the risk of RP has not been clarified. Immune checkpoint inhibitor treatment used during or after chemotherapy increases the risk of RP [16]; therefore, this problem must be verified in the future.

This study has several limitations that require further evaluation. First, our sample sizes were relatively small in both groups, and the subjects were enrolled from a single institution. This was also a retrospective study. A prospective multicenter study is needed to confirm the presented results. Second, in our previous study [9], we adjusted our analysis for ILD and chemotherapy, but we did not in this study. The purpose of this study was to evaluate the best

DVH parameters that can be used in the real-world setting; therefore, we simply compared the threshold of DVH parameters for the prediction of RP. Another clinical element separate from DVH parameters may affect the onset of RP. For example, chemotherapy and ILD were significantly associated with the onset of symptomatic RP in lung cancer (Table 1). We also believe that it is necessary to evaluate these parameters in other malignancies, such as breast cancer and mediastinal tumors. Inspiration-breath-hold scans commonly used for SBRT lung radiotherapy treatments would have larger LAV%, and hence, larger differentials between the segmented and non-segmented metrics. The number of patients who have undergone IMRT is small. This can potentially introduce bias since static IMRT or VMAT plans can have higher, if not much higher, low dose contribution to the treatment. In other words, if the portion of the IMRT plan increases, we may see the optimal dose evaluation criteria shift toward a low dose. The 75% percentiles of HAV20% and V20 and V5 in esophageal cancer were lower than the threshold; therefore, comparing these may not be of much significance.

## Conclusions

DVH parameters with a high attenuation area may have higher commonality than conventional DVH parameters in lung cancer and in esophageal cancer populations. HAV30% may be a better DVH parameter for predicting RP than other conventional parameters.

## Supporting information

**S1 Table. The following dosimetric parameters were evaluated.**
(DOCX)

**S2 Table. Chemotherapy regimens used in the study.**
(DOCX)

**S3 Table. Univariate logistic regression analysis of dosimetric parameters for symptomatic radiation pneumonitis ($\geq$ Grade 2) in lung cancer and esophageal cancer.**
(DOCX)

**S1 Fig.**
(TIF)

## Author Contributions

**Conceptualization:** Yasuki Uchida.

**Data curation:** Yasuki Uchida, Takuya Tsugawa, Kazuo Noma, Ken Aoki, Kentaro Fukunaga, Hiroaki Nakagawa, Masafumi Yamaguchi, Makoto Osawa, Taishi Nagao, Yasutaka Nakano.

**Formal analysis:** Yasuki Uchida, Sachiko Tanaka-Mizuno, Daisuke Kinose.

**Funding acquisition:** Yasuki Uchida, Yasutaka Nakano.

**Investigation:** Yasuki Uchida, Takuya Tsugawa, Emiko Ogawa.

**Methodology:** Yasuki Uchida.

**Project administration:** Yasuki Uchida.

**Supervision:** Yasuki Uchida, Yasutaka Nakano.

**Validation:** Yasuki Uchida, Takuya Tsugawa, Sachiko Tanaka-Mizuno.

**Visualization:** Yasuki Uchida.

**Writing – original draft:** Yasuki Uchida.

**Writing – review & editing:** Yasuki Uchida.

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
