## [Decision Letter · Decision Letter 0]

18 Aug 2020

PONE-D-20-14886

Prediction of radiation pneumonitis using dose-volume histogram parameters with high attenuation in two types of cancer: A retrospective study

PLOS ONE

Dear Dr. Nakano,

Thank you for submitting your manuscript to PLOS ONE. After careful consideration, we feel that it has merit but does not fully meet PLOS ONE’s publication criteria as it currently stands. Therefore, we invite you to submit a revised version of the manuscript that addresses the points raised during the review process.

We look forward to receiving your revised manuscript.

Kind regards,

Dandan Zheng, PhD

Academic Editor

PLOS ONE

Journal Requirements:

2. Thank you for your ethics statement:

"The study protocol was approved by the Institutional Review Board. (IRB no. R2014-236)."

'This work was supported by Daiichi-Sankyo Company, Limited [grant number A19-1287] (YU) https://www.daiichisankyo.co.jp/corporate/ds-shougakukifu/

, Ono Pharmaceutical Company, Limited [grant number ONOS20180618011] (NY)

https://kifu-shinsei.jp/kifu-entry/cmn/doc/index_N1gsZhJaOt.html, Pfizer Inc [grant number 54334665] (NY)

,

https://pfizer-ac-web.pfizer.co.jp/detail.html

Bayer Yakuhin, Limited [BASJ20180409030],

https://byl.bayer.co.jp/researchers/

Boehringer Ingelheim GmbH [grant number RS2019A00769379] (NY)

https://www.boehringer-ingelheim.jp/Research_support_2019

, and Astelas Pharma [grant number RS2019A001313] (NY)

https://www.astellas.com/jp/ja/responsibility/astellas-foundations

and had no role in study design, data collection, data analysis, data interpretation, or writing of the report.'

We note that you received funding from commercial sources.

a. Please provide an amended Competing Interests Statement that explicitly states these commercial funders, along with any other relevant declarations relating to employment, consultancy, patents, products in development, marketed products, etc.

5. Please include captions for your Supporting Information files at the end of your manuscript, and update any in-text citations to match accordingly. Please see our Supporting Information guidelines for more information: http://journals.plos.org/plosone/s/supporting-information

Reviewers' comments:

Reviewer's Responses to Questions

**Comments to the Author**

1. Is the manuscript technically sound, and do the data support the conclusions?

Reviewer #1: No

Reviewer #2: Yes

2. Has the statistical analysis been performed appropriately and rigorously? 

Reviewer #1: No

Reviewer #2: Yes

3. Have the authors made all data underlying the findings in their manuscript fully available?

Reviewer #1: Yes

Reviewer #2: Yes

4. Is the manuscript presented in an intelligible fashion and written in standard English?

Reviewer #1: Yes

Reviewer #2: Yes

5. Review Comments to the Author

Reviewer #1: General comments

The concept of segmenting the lung volume into low density and high density regions, and ignoring the low density is interesting. It is reasonable to expect a differential in the metrics vs clinical outcomes after subtracting out the low density volumes. It does seem, however, that this is a small improvement on current metrics. The paper requires several major additions to more clearly and definitively show that these differences are meaningful.

Major things

Choice of CT number

Please expand on the choice of -856 HU, why this value? could other choices be made?

Ideally one would perform your LAV segmented analysis for a range HU and find the HU number cut-off with the largest improvement in over the non-segmented metrics.

Direct comparison of the of new segmented HAV metrics to corresponding traditional V metrics

The data suggests that the percentage of low attenuation volume is roughly 10% or less of the total lung volume. Doesn’t this mean that the segmentation is only a small (roughly 10% or less) correction compared to standard analyses without the LAV segmentation?

Please directly compare and comment on the statistical significance of the differences between segmented and non-segmented metrics g.g. HAV20% = 33.7 and V20% = 35.2 for lung cancer. Total patients is 77. Is this segmented HAV20% really a significant improvement on the V20%? My quick visual comparison suggests that most the “V” metrics give very similar results to their HAV counterparts.

Same also for the ROC results: e.g HAV10% = 0.869 vs V10%=0.849 in the esophageal ROC figure. Is this difference significant?

Important missing technical details

If possible, also give the mass density or electron densities that the -856 HU threshold value corresponds to for your CT scanner (or at least give a density range).

Please specify the dose calculation algorithms used? This may be particularly significant as simpler algorithms such as Varian’s AAA algorithm, is known to overestimate dose to low density lung volumes compared to Boltsmann-solver or Monte-Carlo algorithms such as accuros.

Please specify the breathing phase of the CT scan (inspiration breathhold , free breathing,4$D CT at a given phase etc). Note that you may want to add to the discussion that for inspiration-breathold scans which are commonly used for SBRT lung radiotherapy treatments would have larger LAV% and hence larger differential between the segmented and non-segmented metrics.

Please also discuss the margins and motion management during treatment (presumably no gating, no breathold was used on these treatments?).

Smaller things

line 49

“have set vague cut-off values for”

strongly suggest to remove word “vague” as the metrics themselves are not vague.

Use of “cut-off” is unusual. Suggest to replace “cut-off” with “constraints” as the constraints can be one of a number of metrics such as: volume that gets at least a given dose, mean dose, max dose.

Line 50-51

“lung volume receiving 20 Gy or more (V20%) (30−40%) to reduce the risk of RP”

Terminology as given is confusing: We need both Dose in Gy and volume in % to define the “V” metrics. Normally the first number (eg V20) is the dose in Gy, not % dose nor % volume. E.g. V20 < 30% means the volume that receives at least 20 Gy should be less than 30% of the lung volume. Here by writing V20% you are implying that the 20 refers to a %, where it normally refers to the dose in Gy. Please correct the your “V20%” notation to the more standard notation “V20 <20%” and complete notation defining both the dose and the volume.

For example see the RTOG 0937 protocol where this constraint originated: https://www.rtog.org/ClinicalTrials/ProtocolTable/StudyDetails.aspx?action=openFile&FileID=13697

Line 52

For clarity suggest to explain or expand on “low attenuation” and “high attenuation” straightaway eg consider “volumes of low attenuation in CT imaging” Suggest to also apply to abstract.

Line 52

It is technically wrong to say “areas of low attenuation” you mean “volumes of low attenuation”

Line 80

“using the threshold limit of −856 HU”

Replace with

“using the upper threshold limit of −856 HU”

Line 106 please specify more clearly and consistent with standard notation

Replace “V2%, V5%, V10%, V20%, and V30%,”

with presumed meaning: “V20 Gy < 2%, 5%, 10%, 20%, 30%”

Please also clearly specify that it is the volume of both lungs if that is the case.

Tables

I find the extreme-spaced tables impossible to digest. E.g table 1 is only 3 data columns but spans 3 pages.

data too spread out to meaningfully digest.

Please specify the resolution, or at least range of resolutions / slice thickness settings used for the CT scans. E.g. voxel size can effect the segmented volumes if the resolution is too course. e.g a voxel that is 4/5 with air 1/5 dense tumor tissue will represented by approx -800HU

Reviewer #2: The manuscript was well written and the analysis was properly done to support the conclusion.

There are a few suggestions for author to consider:

1. It's not very clear in the manuscript how the MU threshold of “- 856” was determined. Was there any systematic analysis done? As you have presented in the manuscript, by setting such a threshold it will impact the selection of optimal dose evaluation criteria and corresponding volume threshold. It's probably a good idea to address this issue to ensure that the difference in dose evaluation was not caused by the uncertainty of the chosen threshold.

2. As what I have noticed that only a small portion of plans were done with IMRT technique, this can potentially introduce bias since static IMRT or VMAT plans can have higher, if not much higher, low dose contribution to the treatment. That means if the portion of the IMRT plan increases, one may see the optimal dose evaluation criteria shift towards low dose.

3. For conventional lung radiation therapy, I can imagine the fractional dose would be similar. However, it would be clearer to include the fractional dose information rather than just the total dose > 30Gy when demonstrating the patient selection criteria.

6. PLOS authors have the option to publish the peer review history of their article (what does this mean?). If published, this will include your full peer review and any attached files.

Reviewer #1: No

Reviewer #2: No

---

## [Author Response · Author response to Decision Letter 0]

2 Oct 2020

Response to Reviewer #1:

Thank you very much for providing us with important insights. We are delighted to hear that you think our work will spark debate in our field. In the following sections, you will find our responses to each of your points and suggestions. We are grateful for the time and energy you expended in reviewing this work.

Major things

Choice of CT number

Please expand on the choice of -856 HU, why this value? could other choices be made?

Ideally one would perform your LAV segmented analysis for a range HU and find the HU number cut-off with the largest improvement in over the non-segmented metrics.

Response:

In general, most studies on COPD have used a CT cut-off value of -950 HU for inspiration and -856 HU for expiration as the cut-off value for emphysematous lesions. Because the regions suggesting an emphysematous lesion in the free-breathing CT images used in this study is close to the images taken during expiration, we used -856HU, which is commonly used during expiration.

To further verify the validity of this cutoff value, we previously validated the association between CT under free breathing and the inspiratory CT performed within 45 days after free breathing CT. Inspiratory CT was performed using Toshiba Aquillion ONE (Toshiba Medical Systems Corp., Otawara, Tochigi, Japan) and LAV was analyzed using the Aquarius iNtuitionTM software ver.4.4.12 (TeraRecon Inc., San Mateo, Calif) and evaluated using the threshold limit of -950 HU. We found that the LAV in inspiratory CT was highly correlated with the LAV in CT under free breathing. This detail can be found in reference #9. We've added these details on lines 87-91. 

Direct comparison of the of new segmented HAV metrics to corresponding traditional V metrics

The data suggests that the percentage of low attenuation volume is roughly 10% or less of the total lung volume. Doesn’t this mean that the segmentation is only a small (roughly 10% or less) correction compared to standard analyses without the LAV segmentation?

Response:

This correction may seem trivial. However, there were differences between individual patients, even if the overall rate was 10%. Some patients had more emphysematous lesions and others had less, and differences appeared between those with larger and smaller corrections. 

An error in the unit of values for the lung cancer population has been corrected.

Please directly compare and comment on the statistical significance of the differences between segmented and non-segmented metrics g.g. HAV20% = 33.7 and V20% = 35.2 for lung cancer. Total patients is 77. Is this segmented HAV20% really a significant improvement on the V20%? My quick visual comparison suggests that most the “V” metrics give very similar results to their HAV counterparts.

Same also for the ROC results: e.g HAV10% = 0.869 vs V10%=0.849 in the esophageal ROC figure. Is this difference significant?

Response:

We agree that you are making a fair point. The AUC was used as a measure of accuracy in this study. In our previous study, we have assessed the AUC, as well as other indices such as NRI (net reclassification improvement) and IDI (integrated discrimination improvement). Although we did not present significant differences in individual values, the main purpose of this study was to compare cut-off values from a highly accurate index to ascertain universality. The purpose was not to compare significant differences in individual values.

Important missing technical details

If possible, also give the mass density or electron densities that the -856 HU threshold value corresponds to for your CT scanner (or at least give a density range).

Response:

Regarding relative electron density, −856HU was 0.1488. We have added these details on lines 92-93.

Please specify the dose calculation algorithms used? This may be particularly significant as simpler algorithms such as Varian’s AAA algorithm, is known to overestimate dose to low density lung volumes compared to Boltsmann-solver or Monte-Carlo algorithms such as accuros.

Response:

correction-based algorithm AAA (version 13.6.23) and the calculation grid was 2.5 mm for the lung. (lines 80-81)

Please specify the breathing phase of the CT scan (inspiration breathhold , free breathing,4$D CT at a given phase etc). Note that you may want to add to the discussion that for inspiration-breathold scans which are commonly used for SBRT lung radiotherapy treatments would have larger LAV% and hence larger differential between the segmented and non-segmented metrics.

Response:

We have added it on line 84 and 244-246.

Please also discuss the margins and motion management during treatment (presumably no gating, no breathold was used on these treatments?).

Response:

Gating and breath-hold were not used, as you pointed out. We have included this information in the manuscript.

Smaller things

line 49

“have set vague cut-off values for”

strongly suggest to remove word “vague” as the metrics themselves are not vague.

Use of “cut-off” is unusual. Suggest to replace “cut-off” with “constraints” as the constraints can be one of a number of metrics such as: volume that gets at least a given dose, mean dose, max dose.

Response:

We have removed the word “vague” and replaced “cut-off” with “constraints.”

Line 50-51

“lung volume receiving 20 Gy or more (V20%) (30−40%) to reduce the risk of RP”

Terminology as given is confusing: We need both Dose in Gy and volume in % to define the “V” metrics. Normally the first number (eg V20) is the dose in Gy, not % dose nor % volume. E.g. V20 < 30% means the volume that receives at least 20 Gy should be less than 30% of the lung volume. Here by writing V20% you are implying that the 20 refers to a %, where it normally refers to the dose in Gy. Please correct the your “V20%” notation to the more standard notation “V20 <20%” and complete notation defining both the dose and the volume.

For example see the RTOG 0937 protocol where this constraint originated: https://www.rtog.org/ClinicalTrials/ProtocolTable/StudyDetails.aspx?action=openFile&FileID=13697

Response:

Thank you for your comment. We have removed "%."

Line 52

For clarity suggest to explain or expand on “low attenuation” and “high attenuation” straightaway eg consider “volumes of low attenuation in CT imaging” Suggest to also apply to abstract.

Response:

Thank you for pointing this out. Your recommendation has been followed, the abstract inclusive.

Line 52

It is technically wrong to say “areas of low attenuation” you mean “volumes of low attenuation”

Response:

We have replaced with "volumes," as suggested.

Line 80

“using the threshold limit of −856 HU”

Replace with

“using the upper threshold limit of −856 HU”

Response:

Following your recommendation, we have added "upper" to the phrase.

Line 106 please specify more clearly and consistent with standard notation

Replace “V2%, V5%, V10%, V20%, and V30%,”

with presumed meaning: “V20 Gy < 2%, 5%, 10%, 20%, 30%”

Please also clearly specify that it is the volume of both lungs if that is the case.

Response:

Thank you for your comment. We have removed "%."

Tables

I find the extreme-spaced tables impossible to digest. E.g table 1 is only 3 data columns but spans 3 pages.

data too spread out to meaningfully digest.

Response:

Thank you for your comment. We have revised the tables for clarity.

Please specify the resolution, or at least range of resolutions / slice thickness settings used for the CT scans. E.g. voxel size can effect the segmented volumes if the resolution is too course. e.g a voxel that is 4/5 with air 1/5 dense tumor tissue will represented by approx -800HU

Response:

Thank you for your comment. We have included this information on lines 82-82.

Reviewer #2: The manuscript was well written and the analysis was properly done to support the conclusion.

There are a few suggestions for author to consider:

1. It's not very clear in the manuscript how the MU threshold of “- 856” was determined. Was there any systematic analysis done? As you have presented in the manuscript, by setting such a threshold it will impact the selection of optimal dose evaluation criteria and corresponding volume threshold. It's probably a good idea to address this issue to ensure that the difference in dose evaluation was not caused by the uncertainty of the chosen threshold.

Response:

In general, most studies on COPD have used a CT cut-off value of -950 HU for inspiration and -856 HU for expiration as the cut-off value for emphysematous lesions. Because the regions suggesting an emphysematous lesion in the free-breathing CT images used in this study is close to the images taken during expiration, we used -856HU, which is commonly used during expiration.

To further verify the validity of this cutoff value, we previously validated the association between CT under free breathing and the inspiratory CT performed within 45 days after free breathing CT. Inspiratory CT was performed using Toshiba Aquillion ONE (Toshiba Medical Systems Corp., Otawara, Tochigi, Japan) and LAV was analyzed using the Aquarius iNtuitionTM software ver.4.4.12 (TeraRecon Inc., San Mateo, Calif) and evaluated using the threshold limit of -950 HU. We found that the LAV in inspiratory CT was highly correlated with the LAV in CT under free breathing. This detail can be found in reference #9.

2. As what I have noticed that only a small portion of plans were done with IMRT technique, this can potentially introduce bias since static IMRT or VMAT plans can have higher, if not much higher, low dose contribution to the treatment. That means if the portion of the IMRT plan increases, one may see the optimal dose evaluation criteria shift towards low dose.

Response:

Thank you for pointing this out. We have included information regarding this on lines 246-249

3. For conventional lung radiation therapy, I can imagine the fractional dose would be similar. However, it would be clearer to include the fractional dose information rather than just the total dose > 30Gy when demonstrating the patient selection criteria.

Response:

The fraction dose is 1.8–3.0 Gy. We have included this in the manuscri

---

## [Decision Letter · Decision Letter 1]

29 Oct 2020

PONE-D-20-14886R1

Prediction of radiation pneumonitis using dose-volume histogram parameters with high attenuation in two types of cancer: A retrospective study

PLOS ONE

Dear Dr. Nakano,

Thank you for submitting your manuscript to PLOS ONE. After careful consideration, we feel that it has merit but does not fully meet PLOS ONE’s publication criteria as it currently stands. Therefore, we invite you to submit a revised version of the manuscript that addresses the points raised during the review process.

ACADEMIC EDITOR: The revision is much improved with only minor issues remaining. Please revise accordingly.

We look forward to receiving your revised manuscript.

Kind regards,

Dandan Zheng, PhD

Academic Editor

PLOS ONE

Reviewers' comments:

Reviewer's Responses to Questions

**Comments to the Author**

1. If the authors have adequately addressed your comments raised in a previous round of review and you feel that this manuscript is now acceptable for publication, you may indicate that here to bypass the “Comments to the Author” section, enter your conflict of interest statement in the “Confidential to Editor” section, and submit your "Accept" recommendation.

Reviewer #1: All comments have been addressed

Reviewer #2: All comments have been addressed

2. Is the manuscript technically sound, and do the data support the conclusions?

Reviewer #1: Yes

Reviewer #2: Yes

3. Has the statistical analysis been performed appropriately and rigorously? 

Reviewer #1: Yes

Reviewer #2: Yes

4. Have the authors made all data underlying the findings in their manuscript fully available?

Reviewer #1: Yes

Reviewer #2: No

5. Is the manuscript presented in an intelligible fashion and written in standard English?

Reviewer #1: Yes

Reviewer #2: Yes

6. Review Comments to the Author

Reviewer #1: The revised version of the manuscript fully addresses all my concerns. Thank you for your corrections.

Reviewer #2: Line 225 - This suggestion may not be appropriate for two reasons: 1. As you have mentioned HAV30% is not the most sensitive predictor of RP for esophagus cases, therefore it’s necessary to have multiple/other predictors in order to reach a useful sensitivity. 2. HAV30% is not the most sensitive predictor for a reason. It’s unlikely that HAV30%/V30 from esophagus will be as high as that in lung cases simply because of the anatomies and beam arrangements (i.e. unlikely you will see an APPA treatment for esophagus as what you have shown as an example for lung). Therefore, the onset of RP could be dominated by the low dose to lung for esophagus cases, which was suggested by the shift of most sensitive predictors from HAV30% in lung to HAV10% and HAV5% in esophagus cases. Therefore, suggesting using HAV30% for both sites can be misleading and may not be clinically useful.

I would also suggest author to generate bar plots of each DVH parameter side by side for the two sites so that it’s clear how the distribution of those parameters different from each other, and compare the calculated thresholds with the 75% percentile of each parameter to see if using that specific predictor is meaningful. For example if the 75% percentile of HAV30% for esophagus cases is 5% and the HAV30% threshold is 11%, it would not be useful to suggest HAV30% as a RP predictor for esophagus cases.

7. PLOS authors have the option to publish the peer review history of their article (what does this mean?). If published, this will include your full peer review and any attached files.

Reviewer #1: No

Reviewer #2: No

---

## [Author Response · Author response to Decision Letter 1]

24 Nov 2020

Response to Reviewer #2:

Thank you very much for providing us with important insights. We are delighted to hear that you think our work will spark debate in our field. In the following sections, you will find our responses to each of your points and suggestions. We are grateful for the time and energy you expended in reviewing this work.

Line 225 - This suggestion may not be appropriate for two reasons: 1. As you have mentioned HAV30% is not the most sensitive predictor of RP for esophagus cases, therefore it’s necessary to have multiple/other predictors in order to reach a useful sensitivity. 2. HAV30% is not the most sensitive predictor for a reason. It’s unlikely that

HAV30%/V30 from esophagus will be as high as that in lung cases simply because of the anatomies and beam arrangements (i.e. unlikely you will see an APPA treatment for esophagus as what you have shown as an example for lung). Therefore, the onset of RP could be dominated by the low dose to lung for esophagus cases, which was suggested by the shift of most sensitive predictors from HAV30% in lung to HAV10% and HAV5% in esophagus cases. Therefore, suggesting using HAV30% for both sites can be misleading and may not be clinically useful.

Response:

We deleted this sentence.

I would also suggest author to generate bar plots of each DVH parameter side by side for the two sites so that it’s clear how the distribution of those parameters different from each other, and compare the calculated thresholds with the 75% percentile of each parameter to see if using that specific predictor is meaningful. For example if the 75% percentile of

HAV30% for esophagus cases is 5% and the HAV30% threshold is 11%, it would not be useful to suggest HAV30% as a RP predictor for esophagus cases.

Response:

We created box plots instead of bar blots to clarify the 75% percentile (S4 Figure). As you pointed out, the 75% percentile of HAV20% and V20 and V5 in esophageal cancer were lower than threshold. Therefor this point was added as the Limitation to Discussion.

---

## [Editor Report · Decision Letter 2]

4 Dec 2020

Prediction of radiation pneumonitis using dose-volume histogram parameters with high attenuation in two types of cancer: A retrospective study

PONE-D-20-14886R2

Dear Dr. Nakano,

We’re pleased to inform you that your manuscript has been judged scientifically suitable for publication and will be formally accepted for publication once it meets all outstanding technical requirements.

Kind regards,

Dandan Zheng, PhD

Academic Editor

PLOS ONE

---

## [Editor Report · Acceptance letter]

8 Dec 2020

PONE-D-20-14886R2 

Prediction of radiation pneumonitis using dose-volume histogram parameters with high attenuation in two types of cancer: A retrospective study 

Dear Dr. Nakano:

I'm pleased to inform you that your manuscript has been deemed suitable for publication in PLOS ONE. Congratulations! Your manuscript is now with our production department. 

Kind regards, 

on behalf of

Dr. Dandan Zheng 

Academic Editor

PLOS ONE